# Sophisticated Conversations between Chromatin and Chromatin Remodelers, and Dissonances in Cancer

**DOI:** 10.3390/ijms22115578

**Published:** 2021-05-25

**Authors:** Cedric R. Clapier

**Affiliations:** Department of Oncological Sciences & Howard Hughes Medical Institute, Huntsman Cancer Institute, University of Utah School of Medicine, 2000 Circle of Hope, Salt Lake City, UT 84112, USA; cedric.clapier@hci.utah.edu

**Keywords:** chromatin remodeling, ISWI, CHD, SWI/SNF, INO80, BAF, nucleosome positioning, transcription regulation, promoter, cancer

## Abstract

The establishment and maintenance of genome packaging into chromatin contribute to define specific cellular identity and function. Dynamic regulation of chromatin organization and nucleosome positioning are critical to all DNA transactions—in particular, the regulation of gene expression—and involve the cooperative action of sequence-specific DNA-binding factors, histone modifying enzymes, and remodelers. Remodelers are molecular machines that generate various chromatin landscapes, adjust nucleosome positioning, and alter DNA accessibility by using ATP binding and hydrolysis to perform DNA translocation, which is highly regulated through sophisticated structural and functional conversations with nucleosomes. In this review, I first present the functional and structural diversity of remodelers, while emphasizing the basic mechanism of DNA translocation, the common regulatory aspects, and the hand-in-hand progressive increase in complexity of the regulatory conversations between remodelers and nucleosomes that accompanies the increase in challenges of remodeling processes. Next, I examine how, through nucleosome positioning, remodelers guide the regulation of gene expression. Finally, I explore various aspects of how alterations/mutations in remodelers introduce dissonance into the conversations between remodelers and nucleosomes, modify chromatin organization, and contribute to oncogenesis.

## 1. The Nucleosome, Substrate of the Remodelers

### 1.1. Composition and Stability of the Nucleosome

The canonical nucleosome, the fundamental repeating unit of chromatin, consists of a histone octamer, associating two H3/H4 dimers forming a central tetramer capped on each side by an H2A/H2B dimer, around which 147 bp of DNA wrapped in ~1.7 tight left-handed superhelical turns [1]. Within a nucleosome, there are fourteen histone–DNA contacts established between positively charged amino acids from the histones and negatively charged phosphate backbones of the DNA, three per histone dimer, and two weaker contacts at the entry/exit to the nucleosome provided by extensions from each histone H3. While each contact is relatively weak (~1 kcal/mole), requiring ~1 pN of force to be disrupted, all fourteen contacts together (~12–14 kcal) confer significant positional stability. By convention, DNA locations within the nucleosome are described as superhelical locations (SHL) followed by a digit (from the DNA entry to exit sides: +7 to +1, 0, −1 to −7) corresponding to each successive major groove facing the histones with respect to SHL0 or dyad which is the location of the central base pair of the DNA where the histone octamer pseudo-symmetry axis intersects DNA. Thus, locations of remodeler components that interact with the nucleosome are commonly described by using SHL numbering (applied below).

Despite being the fundamental repeating units of chromatin, nucleosomes contribute very modestly to the physical organization and packaging of the genome in the nucleus (introducing only 7-fold compaction to the DNA). Instead, nucleosomes confer major functional organization to the genome by altering DNA accessibility and excellent support to regulatory information [2]. Indeed, conceptually, nucleosomes compete with DNA-binding proteins to occupy sites in the genome and provide selective access to DNA, as a majority of DNA-binding factors are prevented from binding to their cognate sites if wrapped into nucleosomes. To manipulate this obstacle and provide plasticity to chromatin, remodelers have to overcome at least part of the positional stability of the nucleosome.

While the structure of the nucleosome is largely unaffected by the DNA sequence, the positional stability of the histone octamer depends on the underlying sequence of the DNA. Indeed, the DNA sequence affects DNA stiffness and curvature, thus its bending tolerance and its susceptibility to accommodate the curvature imposed by the wrapping around the octamer. For example, the increased stiffness of the AT-rich DNA sequences renders nucleosomes positioned over those sequences less stable [3,4]. Of note, DNA methylation also reduces DNA flexibility, altering nucleosome positioning and stability [5,6]. Thus, the DNA sequence influences the organization of chromatin in general, and at enhancers and promoters in particular. Intuitively, promoter DNA sequences that increase nucleosome stability result in decreased transcription [7], and the action of remodelers can either be supported or limited by this variability in nucleosome stability and positioning (see Section 1.2).

Finally, the positioning and stability of a nucleosome are also modified by the presence of histone variants, histone post-translational modifications (PTMs), high-mobility-group (HMG) proteins, or linker histones in specialized chromatin regions, adding to the features susceptible to affect remodelers’ activities (reviewed in Reference [8]; see Section 4.5.6).

### 1.2. Nucleosome Positioning, Spacing, and Phasing, and the Roles of Remodelers

Remodelers can be classified into categories specialized in nucleosome editing, chromatin assembly, or chromatin opening (Figure 1A); thus, they can participate differently in nucleosome organization. The position of a nucleosome is established randomly during the histone deposition process, which occurs mainly during replication, and the positional stability of a nucleosome depends on the features described above. Thus, in a population of genomes (right after replication), DNA-binding sites are randomly exposed between nucleosomes or hidden within a nucleosome (Figure 1B(1)). By moving octamers in *cis* (Sliding) or in *trans* (Ejection), remodelers actively contribute to nucleosome positioning and play a critical role in regulating accessibility for DNA-binding factors. In addition, nucleosomes are distributed in arrays along the DNA, and the distance separating them defines nucleosome spacing. In a population of genomes, from the initial randomly deposited octamer, assembly remodelers can achieve regular spacing of the nucleosomes at a given locus. Independently, they however cannot achieve phasing, the uniform nucleosome positioning relative to the underlying DNA sequence in a population of genomes, because the position of the reference nucleosome may differ between genomes (Figure 1B(2)). Phasing can result from uniform nucleosome positioning being dictated by obstacles such as the presence of a barrier factor (e.g., transcription factor and boundary element), constraining an assembly remodeler to uniformly space nucleosomes with respect to the obstacle, resulting in regular spacing with phasing across genomes (Figure 1B(3)). Furthermore, from a regularly spaced and phased population, opening remodelers can remove nucleosomes, leading to a population of phased but not evenly spaced nucleosomes, a situation common at promoters in order to homogeneously grant activators access to their cognate sites (Figure 1B(4)). Consequently, uniform exposure of DNA-binding sites results from arrays that are spaced and phased. These aspects are critical to appreciate the organization of chromatin, particularly at promoters and enhancers, and the contribution of remodelers to transcription regulation (see Section 5).

Rather than being intrinsically DNA-encoded or transcription-related, in vivo nucleosome positioning and the highly regularly spaced and phased nucleosomal distribution near the 5′ends of most eukaryotic genes relies instead on ATP-dependent remodelers stacking nucleosomes against a barrier such as CTCF [9,10]. Remarkably, in yeast, an elegant study measuring the intrinsic cyclizability of DNA fragments demonstrated that the nucleosome-depleted regions (NDRs) upstream to the transcription start sites (TSS) possess unusually low bendability that inhibits INO80-mediated nucleosome sliding into the NDR, likely defining a mechanical barrier [11].

## 2. Functional Classification and Specialization of Remodelers

### 2.1. Chromatin Remodeling Outcomes and Classification of Remodelers

To manipulate the nucleosomal barrier and assist selective access to the DNA, chromatin remodelers use the binding and hydrolysis of ATP molecules to break DNA–histone contacts in the nucleosome, leading to a variety of outcomes that can be classified into three categories (Figure 1A): (1) nucleosome editing in which the composition of the nucleosome is modified by the installation or removal of histone variants, (2) chromatin assembly in which nucleosomal DNA is tightly wrapped (by maturation of initially deposited histone–DNA complexes into canonical nucleosomes) and *cis* displacement of the histone octamer (sliding) results in regularly spaced arrays, and (3) chromatin opening in which either irregularly spaced arrays are generated by sliding or nucleosome wrapped DNA becomes more accessible due to ejection of the entire histone octamer, partial disassembly of the nucleosome with an H2A/H2B dimer eviction, or altered DNA wrapping.

In parallel, chromatin remodelers are traditionally classified into four major subfamilies named after their core ATPases, namely ISWI, CHD, SWI/SNF, and INO80, which are separated based on similarities and differences between domains harbored within their ATPase subunits (Figure 2A). The vast majority of eukaryotes contains at least one remodeler from each of the four subfamilies, supporting the notion that their functions are largely non-redundant. Moreover, higher eukaryotes have evolved a broader set of remodelers within each subfamily by increasing compositional diversity through a modular and combinatorial architecture involving subunit paralogs, leading to functionally tailored remodelers, such as cell-type specific or developmentally specific remodelers (reviewed in Reference [12]).

### 2.2. Specialization of the Remodelers

#### 2.2.1. ISWI and CHD Subfamilies

ISWI-subfamily remodelers generally facilitate chromatin assembly. Notably, with histone chaperones, they assist the maturation of histone–DNA complexes (prenucleosomes) into canonical nucleosome and proper density and spacing during replication, defining the initial chromatin landscape [16,17,18]. Their central function is to generate evenly spaced nucleosomes of various compositions in different chromatin contexts, optimizing the packaging of subsections of the genome (Figure 1B(2)). When coordinated with obstacles such as barrier factors (boundary elements or transcription factors), some assembly remodelers generate spaced and phased arrays, leading to a homogeneous site blockage or exposure to site-specific DNA-binding factors (Figure 1B(3)). In order to evenly space nucleosomes, the ISWI and CHD subfamilies’ motor subunits harbor a DNA-binding domain (DBD) or a HAND–SANT–SLIDE (HSS) domain, which is used as a molecular ruler, and display interaction with and regulation by H4 tails (Figure 3A,B) (see Section 4.5.2 and Section 4.5.3) [19,20,21,22,23,24,25]. In contrast, a subset of ISWI-subfamily remodelers, which includes the Nucleosome Remodeling Factor (NURF) complex, is involved in increasing DNA accessibility and promoting transcription. The diversity of ISWI-subfamily remodelers has been substantially expanded in evolution, most notably by the incorporation of two alternative motor subunits (SNF2L and SNF2H) along with many alternative regulatory subunits, revealing an increased functional complexity and versatility ripe for further investigation [26].

Although Chd1, the flagship of the CHD subfamily of remodelers, facilitates chromatin assembly, the CHD ATPases themselves are quite diverse, and, depending on their associated subunits and the chromatin context, this subfamily of remodelers can contribute to all outcomes: chromatin assembly [27], chromatin opening [28], and nucleosome editing [29,30]. A prominent example is the Nucleosome Remodeling Deacetylase (NuRD) complex, which associates DNA translocation by CHD3/4 to histone deacetylation through a dynamic and modular architecture [31,32]. Finally, the various CHD-subfamily ATPases themselves can exhibit distinct nucleosome binding and remodeling activities, as exemplified by CHD6, CHD7, and CHD8 [33].

#### 2.2.2. SWI/SNF Subfamily

SWI/SNF-subfamily remodelers enable chromatin opening needed for all DNA transactions, particularly by exposing binding sites to transcriptional activators or repressors. A specific feature of the SWI/SNF-subfamily remodelers is their capacity to expose large regions of DNA by performing nucleosome ejection, guided by transcription factors and influenced by the strength of the nucleosome positioning due to the underlying DNA sequence and histone composition. In yeast, the Remodels the Structure of Chromatin (RSC) complex, has served as a model for understanding how nucleosome ejection versus sliding are chosen and conducted. Interestingly, the choice of remodeling outcome relies on the regulation of the DNA translocation efficiency by the conserved and essential Actin Related Proteins (ARP), which are bound to the HSA domain of the ATPase, and a regulatory hub that physically associates the domains flanking and separating the ATPase lobes [14,34,35] (see Section 4.6). Notably, nucleosome ejection might also occur by spooling DNA off of the adjacent nucleosome via DNA translocation [36,37,38,39]. Here, future work will reveal how the chromatin context and specific remodeler features drive the choice between sliding and ejection, or enable spooling.

In contrast to the ISWI and CHD subfamily remodelers, which are more often small complexes, the SWI/SNF subfamily remodelers are uniformly large multi-protein complexes with conserved modular subunit organization and nucleosome recognition mechanism across complexes and organisms (Figure 3C). The variety of modules is expanded by the incorporation of paralogs, and further increases with the organism’s cellular and developmental diversity and complexity of differentiation. This organizational modularity related to functional versatility was first identified and characterized in yeast, flies, and mammals, and corroborated in all the recently determined structures of SWI/SNF-subfamily remodelers, including the human BAF remodelers [40]. Here, the field would benefit from additional functional characterization of the various SWI/SNF accessory subunits, and future work will need to precisely associate SWI/SNF remodelers of particular compositions to specific functional contexts, such as cell self-renewal, maintenance of cellular identity and highly specialized function, precise transcriptional switching, distinct cellular commitment during development and differentiation.

#### 2.2.3. INO80 Subfamily

INO80-subfamily remodelers mainly edit nucleosome composition by exchanging histone variants, increasing the capacity to specialize one or few nucleosomes locally or temporally, potentially changing the nucleosome stability and/or the recognition by other chromatin factors.

One prominent feature common to all the INO80 subfamily of remodelers, orchestrating their common architecture, is the compositional presence of RuvBL1/2 (in human, Rvb1/2 in yeast) AAA+ ATPases, which assemble into a single ring of hetero-hexamers, bind to the large insertion present in the ATPase domain, and act as an assembly chaperone and scaffold [41,42] (Figure 3D). The presence of this complex enzymatic ring structure in the context of chromatin remodelers remains functionally intriguing.

SWR1 drives the sequential replacement of H2A–H2B dimers with the variant H2A.Z–H2B dimers [43,44], while performing DNA translocation but not causing sliding [45], perhaps by constraining the translocated region within the confines of the nucleosome. Of note, SWR1 is largely, but not completely, unable to slide nucleosomes [46]. In order to achieve the appropriate dimer replacement, SWR1C specifically recognizes H2A.Z–H2B over H2A–H2B via dedicated domains and subunits [47,48]. Mirroring SWR1 nucleosome editing activity, INO80 replaces the variant H2A.Z–H2B dimers with canonical H2A–H2B dimers [49] by performing DNA translocation with higher efficiency in the presence of H2A.Z [50]. Notably, INO80 is also competent in repositioning nucleosomes [51]. Moreover, yeast INO80 can also remove the variant H2A.X [52], and the vertebrate p400 complex can replace H3.1 by H3.3 [53]. Here, shedding light on the mechanisms leading to the high selectivity for specific histone variants and the interplay with histone chaperones is of the utmost interest, with considerable recent progress.

Together, remodelers from all subfamilies contribute to chromatin plasticity by establishing, maintaining, and dynamically tailoring a broad variety of chromatin landscapes necessary for all DNA transactions, including DNA recombination, DNA replication, DNA repair, and transcription.

## 3. Characteristics and Mechanism Shared by All Remodelers

### 3.1. Compositional Characteristics Shared by All Remodelers

Despite their functional diversity, all remodelers share common features (Figure 2C):(1)Remodelers all contain a single catalytic subunit harboring an ATPase domain split into two RecA-like lobes, functioning as a DNA translocase, breaking histone–DNA contacts in nucleosomes.(2)Remodelers all present domains and/or subunits that regulate the ATPase domain and the translocation process, displacing DNA relative to the octamer.(3)Remodelers all exhibit domains that bind histones and specific post-translational modifications, resulting in a greater affinity for the nucleosome than free DNA, and imparting the potential for regulation of remodeling by those modifications.(4)Remodelers all contain domains and/or subunits that can bind free/extranucleosomal DNA, which can inform the remodeling process.(5)Remodelers all include domains and/or subunits that interact with other chromatin factors, e.g., histone chaperones or transcription factors, which can contribute to targeting.

Together, these shared aspects allow the selective engagement and regulated action of specific remodelers on distinct nucleosomes within distinct functional contexts.

### 3.2. DNA Translocation, the Underlying Mechanism Shared by All Remodelers

All remodelers contain a single catalytic subunit belonging to the SF2 superfamily of helicases and DNA/RNA translocases, characterized by an ATPase domain split into two domains, termed lobes 1 and 2, with homology to the *E. coli* RecA DNA-binding protein. The cleft separating the two lobes can host an ATP molecule and the DNA. Supported by a plethora of evidence from computer simulation, and structural and functional observations, the most recent model of DNA translocation, which is suggested to be common to all remodelers, proposes that, during any nucleosome remodeling event, the motor subunit undergoes a conformational cycle driven by the binding and hydrolysis of ATP in the cleft concomitant to an alternating high affinity of the two lobes for the DNA tracking strand [15,54,55,56] (Figure 2B). An ATP molecule binds into a binding pocket formed mainly by the Walker A and Walker B motifs present in lobe 1 when the enzyme is in an open conformation, and requires for its hydrolysis the closing of the cleft and an arginine finger present on lobe 2. The binding and hydrolysis of ATP alternatively reduces the affinity of lobe 1 and lobe 2 for the DNA tracking strand which, combined with the closing–opening cycle of the cleft, leads to the progression of the lobes along the DNA tracking strand in a 3′ to 5′ direction, using an inchworming mechanism with a 1 bp step per ATP cycle [57,58,59,60,61,62]. However, in the context of nucleosome remodeling, the lobes are prevented from moving along the nucleosomal DNA path by being tethered/anchored to the nucleosome through contacts established between the remodelers’ additional domains/subunits and nucleosomal features (i.e., histone acidic patch, histone H4 tails, post-translational modifications, opposite DNA gyre, and extranucleosomal DNA; see Section 4.5). Thus, instead of moving the remodeler along the nucleosomal DNA path, the inchworm mechanism of the lobes translocates the DNA unidirectionally at the surface of the histone octamer, creating a ratcheting cycle of small (1 bp/ATP) DNA deformations, creating DNA translocation and twist defects that pull the DNA toward the octamer dyad while breaking histone–DNA contacts in the nucleosome [57,63,64]. In support, the different nucleotide-bound states revealed by the structures support a model of DNA translocation in which a localized DNA distortion introduced, for example, by the Snf2 remodeler, propagates across the surface of the octamers in small coordinated movements [65]. A DNA distortion at SHL2 is also observed when SWR1 interacts with the nucleosome in an ATP-bound closed conformation [66]. Notably, the power stroke appears to result from ATP binding, while the resetting of the remodeler conformation results from ATP hydrolysis [66,67] (Figure 2B).

Notably, during this DNA translocation process, although the histone octamer does not endure major deformation [56], changes have been reported that might accompany the propagation of the DNA translocation and twist defects [68,69,70]. In addition, the histone core undergoes a distortion resulting from conformational changes at the interface between H2A–H2B and H3–H4 upon SWR1 interaction that might facilitate histone exchange [66]. Finally, as nucleosomes are slightly flexible regarding the DNA they can accommodate, nucleosome stability might be partly ensured by a buffering of a few base-pairs within the nucleosome, with the movement of the entry-side DNA preceding that of exit-side DNA, for example, during sliding by Chd1 and SNF2H [62].

As a diversity of remodeling outcomes has to arise from a common DNA translocation mechanism applied to akin nucleosomal substrates, similarities and unique specificities in the remodeler–nucleosome interaction and precise regulation must be implemented, governing sophisticated conversations between nucleosomes and remodelers.

## 4. Shared and Specific Regulatory Principles and Features Governing the Sophisticated Conversations between Nucleosomes and Remodelers

### 4.1. Blossoming of Structures with Remodelers Engaging Nucleosomes

Recently, a profusion of structures has been obtained by cryogenic electron microscopy displaying chromatin remodelers engaged with nucleosomes. This has strengthened and expanded the conceptual unification in the mechanistic and regulatory principles and features uncovered previously through genetic, biochemical, and biophysical studies.

To date, the structures of the following nucleosome-bound remodelers have been released: in the ISWI and CHD subfamilies, yeast Chd1 [15], yeast ISW1 [56], human SNF2H [71], and human CHD4 [72]; in the SWI/SNF subfamily, yeast RSC [73,74,75,76], yeast SWI/SNF [77], human recombinant, and endogenous BAF [78,79]; and in the INO80 subfamily, yeast SWR1 [66], *C. thermophilum*, and human INO80 [80,81]. Examples from each of the four subfamilies of remodelers are presented in Figure 3 and commented upon below.

### 4.2. Localization of the Catalytic Core, and Free Versus Restrained Translocation

All remodelers interact with nucleosomes in a multivalent fashion (Figure 3). The main interaction shared by all remodelers corresponds to the binding of the two ATPase lobes to the DNA. The structures confirmed previous work demonstrating that a vast majority of the remodelers’ lobes engage the DNA within the nucleosome at the SHL2 position [57,82,83], with the notable exception of the INO80 lobes located at SHL-6/-7 [50,80,81] (Figure 3).

With the lobes located at SHL2, it is established that, in the process of sliding, the ISWI and CHD families’ remodelers perform DNA translocation iteratively, which freely propagates DNA, breaking a single histone–DNA interaction at a time, whereas, in the process of ejection, the SWI/SNF-family remodelers have the regulated opportunity to perform forceful DNA translocation, breaking multiple histone–DNA interactions simultaneously. In contrast, in the process of histone exchange by SWR1 and INO80 remodelers, the DNA translocation is spatially limited and transient, rupturing few histone–DNA interactions, locally distorting and unwrapping DNA, exposing the histone dimer, and providing an opportunity for dimer exchange while preventing net sliding and preserving nucleosome integrity. Importantly, this restrained DNA translocation is achieved by adding a counter-grip formed by the Arp6/Swc6 module in SWR1 and the Arp5/Ies6 module in INO80 [45,46,66,80,81,84]. Strikingly, SWR1 and INO80 share a common overall architecture but with an inverted organization. Thus, the interactions with the nucleosome are inverted: in SWR1, as the motor is bound at SHL2, the counter-grip Arp6/Swc6 locates at SHL6; meanwhile, in INO80, as the motor is bound at SHL-6, the counter-grip Arp5/Ies6 locates at SHL-2/-3 (Figure 3D). Consequently, in the case of SWR1, as in the ISWI, CHD and SWI/SNF subfamilies of remodelers, the motor pumps DNA from the entry side, but pulls it from the counter-grip; and the ruptured histone–DNA contacts are thus located upstream of the motor. In contrast, in the case of INO80, while the motor still pumps DNA from the entry side, it pushes it against the counter-grip; and the ruptured histone–DNA contacts are thus located downstream of the motor. Of note, in INO80, the stabilization of the remodeler–nucleosome interactions is reinforced by the interaction of lobe 2 with the Rvbs, and by the Ies2 subunit, which is characterized as a critical regulator [85,86,87] and connects lobe 1 to SHL2.

### 4.3. Orchestration of Remodeler Architecture by the ATPase Subunit

Beyond the catalytic core, the ATPase subunit proteins also harbor critical specialization and regulatory domains, such as the HSS/DBD in the ISWI and CHD subfamilies. In addition, in the large remodelers from the SWI/SNF and INO80 subfamilies, the ATPase subunit is the major structural organizer of the complexes (Figure 3C,D). Indeed, in the SWI/SNF-subfamily remodelers, the ATPase subunit itself represents the backbone of the entire remodeler with its N-terminus extending all the way across the whole structure of the complex, along which most of the other components are organized in distinct structural and functional modules, including an HSA-bound ARP module. Toward the N-terminus of the ATPase, multiple subunits form a large base, a substrate recruitment module, located mostly on the distal side of the nucleosome with a massive core scaffold bundling four coiled-coil domains from which three substrate-binding lobes bud: a nucleosome binding lobe, a DNA-binding lobe, and a histone-tail binding lobe, which, together, are involved in substrate selection (detailed in Section 4.4, Section 4.5, Section 4.6, Section 4.7, Section 4.8). The ATPase subunits of the INO80-subfamily remodelers, as those of the SWI/SNF-subfamily remodelers, are highly extended proteins establishing direct contacts with all the modules forming the remodeler, spanning from the catalytic core and an HSA-bound ARP module to various subunits organized in modules involved in anchoring the motor, sensing the substrate, and contacting the acidic patches or the extranucleosomal DNA (detailed in Section 4.4, Section 4.5, Section 4.6). Together, this architecture ensures proper operation and opportunities for fine-tuned and context-dependent regulation via integration of allosteric activations and inhibitions.

### 4.4. Positive Correlation Between Functional, Structural, and Regulatory Complexities Throughout Remodelers

Notably, the complexity of the regulatory conversation between a nucleosome and a remodeler increases with the challenges faced, and the complexity of the processes necessary to achieve various remodeling outcomes (Figure 4 and detailed in Section 4.5, Section 4.6, Section 4.7, Section 4.8). This is in part reflected in increased compositional and structural complexities of the corresponding remodelers in order to be capable of overcoming challenges they face to fulfill their function, and in the multiplicity of their contacts with the nucleosome.

Logically, while all remodelers interact with nucleosomes in a multivalent fashion (Figure 2C), the number, nature, and extent of the interaction increase along with the complexity of the aimed process (Figure 4): in chromatin assembly (left arrow), nucleosome sliding requires minimal action, and thus interaction, spacing, and phasing still require minimal action with slightly more interaction; in chromatin opening (central arrow), nucleosome ejection and the alternative sliding require bold and robust action with advanced regulation guiding the choice of the outcome; in nucleosome editing (right arrow), histone exchange requires significant but spatially restrained action with subtle substrate identification and preservation of the nucleosome integrity. Thus, evidently, the histone exchangers from the INO80 subfamily are the remodelers that present the most extended interactions with the nucleosome and the histones in order to generate a situation that favors dimer exchange while preserving the integrity and stability of the rest of the nucleosome (detailed in Section 4.5 and Section 4.6).

### 4.5. Regulation of Remodelers by Nucleosomal Features

#### 4.5.1. Autoinhibition in Remodelers

A key regulatory principle that appears common to a growing number of remodelers is the presence of intrinsic autoinhibitions in their nucleosome unbound state, which are released to enable catalytic activation upon binding to a nucleosome (Figure 4). Examples abound and include the following: Chd1 is gated by chromodomains [88], ISWI and SNF2H are auto inhibited by AutoN and NegC [89,90], ERCC6/CSB is autoinhibited by its N-terminal region [91], Rhp26 is autoinhibited by a ‘leucine latch’ motif [92], ALC1/CHD1L is autoinhibited by its macro domain [93,94], and now support from structural studies have emerged for ISWI [95] and Snf2 [96]. Thus, consistently, the catalytic cores of the ATPase/DNA translocase subunits are intrinsically active [35,88,89,97], and held in check via autoinhibition.

#### 4.5.2. Regulation by the Histone H4 Tail

Remarkably, the histone H4 tail (which arises from the octamer close to SHL2 position) and its basic patch are required for the proper ATPase and remodeling activities of many ISWI and CHD-subfamily remodelers: the CHRomatin Accessibility Complex (CHRAC) and ISWI [19,98], NURF [20], ISW2 [99], Chd1 [100], and SNF2H [101]. Of note, chromatin compaction and the stability of the interaction between nucleosomes involve the histone H4 tail and depend on its acetylation status [102]. The ISWI remodeler appears to be an intrinsically active DNA translocase held in check by two autoinhibitory regions, one of them being the AutoN region, which displays sequence similarities to the H4 tail basic patch and restrains the ATPase activity of the remodeler [89]. The activation of the ISWI ATPase activity involves the release of the AutoN inhibition by the histone-H4-tail basic patch [89], a regulatory mechanism supported by structural studies of ISWI bound to a nucleosome [56]. Moreover, in the ATP-utilizing Chromatin assembly and remodeling Factor (ACF) remodelers, the inhibited state is reinforced by the interaction of the H4 tail with the N-terminal portion of the accessory subunit Acf1, which is released by its preferential binding to extranucleosomal DNA, resulting in allosteric regulation [103]. Beyond ISWI, full remodeling activities of Snf2, Chd1, CHD4, and SNF2H require the histone H4 tail, which interacts with the conserved acidic cavity located at the surface of lobe 2 [15,65,71,72] (Figure 4).

#### 4.5.3. Regulation by the Presence and Length of Extranucleosomal DNA

Besides AutoN, the DNA translocase activity of the ISWI remodeler is also held in check by a second autoinhibitory region, the NegC region, which prevents the coupling of energy consumption to productive DNA translocation [89]. Remarkably, the establishment of efficient DNA translocation by ISWI involves the release of this NegC inhibition through the binding of the subfamily signature HAND–SANT–SLIDE (HSS) domain to the extranucleosomal DNA [89]. The HSS domain acts as a DNA sensor, a molecular ruler, critical to equally space nucleosomes in arrays, resulting in optimized chromatin assembly and packaging, the function of most ISWI subfamily remodelers. Notably, the HSS domain can be complemented by Acf1 in the ACF remodeler (see Section 4.5.2).

Similarly, the CHD motor subunit contains a NegC domain and a DNA-binding domain (DBD) associating SANT and SLIDE domains, which can be used as a DNA sensor to evenly distance nucleosomes [25]. As in ISWI, the engagement of yeast Chd1 with a nucleosome requires the reorientation of the DBD, lifting an autoinhibitory state [104]. In addition, the closure of the yeast Chd1 ATPase domain depends on a swinging movement of the double chromodomains, a subfamily signature, towards nucleosomal DNA [15], consistent with the physical gating of the DNA binding by the chromodomains [88] (Figure 4).

Consistently, a minimal length of extranucleosomal DNA is necessary for optimal remodeling activities by various remodelers, particularly for members of the ISWI and CHD subfamilies, as they rely on a molecular ruler to space nucleosomes [105,106]. In contrast to Chd1, in which the reorientation of the DBD contributes to unwrapping of the distal DNA, the binding of CHD4 to a nucleosome does not induce DNA unwrapping, an observation consistent with the role of CHD4 in gene repression and heterochromatin formation and maintenance [72] (Figure 3B). Furthermore, the engagement of the RNA Polymerases II (RNAPII) with a nucleosome activates Chd1 by releasing its DBD interaction with extranucleosomal DNA and its contact with the DNA second gyre, leading to the progression of RNAPII through the nucleosome [107,108]. Of note, some of the CHD-subfamily motor subunits harbor additional tandem plant homeodomain (PHD) fingers in their N-termini while others contain a CHD1 helical C-terminal (CHCT) domain in their C-termini [109] that might further contribute to DNA binding.

In SWI/SNF remodelers, from its subunit and domain composition (e.g., HMG in SMARCE1 and Wedge Helix (WH) in SMARCB1), the DNA-binding lobe appears to be poised to bind extranucleosomal exit DNA and recognize promoter DNA elements, possibly contributing to the DNA translocation process.

Notably, the regulation of the sliding activity of the INO80 remodeler also involves sensing the length of the extranucleosomal DNA [110,111]. In SWR1C, the extranucleosomal DNA can be bound by the Arp4–Actin–Arp8–HSA domain, acting as a DNA sensor and regulating activity and coupling [112]. Remarkably, the contacts made by Arp5/Ies6 in INO80 mirror those made by Arp6/Swc6 in SWR1, in particular the tethering to the histone octamer and the interaction with the extranucleosomal entry DNA [66,80,81].

Here, it will be highly conceptually relevant to investigate, in future work, the mechanistic and regulatory impacts of sensing the length of extranucleosomal DNA, its composition/stiffness, and the presence of a barrier factor, particularly at promoters for the regulation of the +1 nucleosome positioning (see Section 1 and Section 5.2).

#### 4.5.4. Regulation by the Histone Acidic Patches

The histone acidic patch is defined by a conserved cluster of eight negatively charged residues from histones H2A and H2B that form a narrow groove at the surface of the nucleosome. The acidic patch of a nucleosome is contacted by the histone H4 tail of another nucleosome in the crystal lattice [1] but can also structurally interact with various chromatin factors: LANA [113], RCC1 [114], Sir3 [115], HMGN2 [116], CENP-C [117], and Ring1B of PRC1 [118]. Confluently, all chromatin remodelers functionally interact with the nucleosomal acidic patches. Indeed, the remodeling activity of remodelers from different subfamilies, ACF, CHD4, and BRG1, is reduced upon neutralization of charges in the acidic patch [119]. Moreover, crosslinking mass spectrometry revealed a structural interaction between the two autoinhibitory regions AutoN and NegC of SNF2H and the acidic patch [90], suggesting a complex regulation and a potential fine-tuning of the release of the autoinhibition in conjunction with the histone H4 tail and the extranucleosomal DNA, discussed above. Notably, the ISWI ATPase contacts the nucleosome acidic-patch via an acidic-patch binding region flanking the HSS domain, a likely anchor that facilitates DNA translocation [120] (Figure 4). Moreover, the direction of sliding by ISWI is impacted by an asymmetry between the two acidic patches [121]. Interestingly, the ALC1/CHD1L remodeler strongly depends on the binding of its linker segment to the proximal acidic patch for tethering and efficient coupling of ATP hydrolysis to nucleosome remodeling [122].

Furthermore, recent structures of SWI/SNF-subfamily remodelers, RSC [73,75], SWI/SNF [77], and BAF [78,79], revealed direct interactions between those remodelers and both acidic patches of the nucleosome. First, the conserved SnAC (Snf2 ATP-coupling) domain, and/or a basic patch immediately following the SnAC domain [76], of the ATPase binds to the histones [123] at the proximal acidic patch, likely to form an anchor contributing to efficient nucleosome sliding (Figure 4). Of note, this interaction might be influenced by the nearby H4 tail [75]. In the meantime, the conserved basic residues of the C-terminal tail, the finger helix, of the essential subunit Sfh1/Snf5/BAF47–SMARCB1 interacts with the distal acidic patch [124], further anchoring the remodeler to the octamer, and promoting nucleosome ejection [73]. Here, this additional anchor might be necessary to increase the torque applied to the DNA by the remodeler, and to achieve the concomitant rupture of several histone–DNA contacts and thus nucleosome ejection (Figure 4).

Moreover, SWR1 and INO80 remodelers functionally interact with both acidic patches [66,80,81]. Here, the structure of SWR1 confirmed that Swc6 establishes contacts with the H2A tail and the proximal acidic patch [45], possibly contributing to the specificity of SWR1 for H2A-containing nucleosomes [44]. Remarkably, the structure also reveals that the Swc2 subunit of SWR1 travels all the way across the nucleosome to interact with the acidic patch of the nucleosome distal face [66]. In INO80, in a stunningly similar situation, the Arp5 subunit contains a grappling insertion that interacts with the proximal acidic patch while the Ies2 subunit stretches across the nucleosome to interact with the distal acidic patch of the nucleosome, as does Swc2 in SWR1, despite no sequence similarities [80,81] (Figure 4). In support of a critical role for those interactions, acidic patches are required for nucleosome sliding by INO80 [80]. Notably, INO80, which acts as a dimer in repositioning nucleosomes [51], might require the Arp5 and Ies2 subunits to compete for the acidic patches on each side of the histone octamer [80].

It is remarkable that the remodelers from the SWI/SNF and INO80 subfamilies establish critical contacts with both nucleosomal acidic patches, contrasting with the remodelers from the ISWI and CHD subfamilies, which interact with only the proximal acidic patch. This observation aligns well with the need for the INO80 and SWI/SNF subfamilies’ remodelers to substantially increase their anchoring contacts with the histones in order to apply stronger torque to the DNA, and thus simultaneously break several histone–DNA contacts, necessary for histone exchange and nucleosome ejection, respectively (Figure 4).

#### 4.5.5. Regulation by the Nucleosomal DNA Sequence, and Contacts to the Second DNA Gyre

Without modifying the remodeling mechanisms per se, nucleosomal DNA sequence-dependent differences have been observed in remodeling activities with many remodelers, likely due to the altered intrinsic stability of the nucleosome. One of the prominent examples is the RSC remodeling activity being stimulated by AT-rich sequences in the nucleosome, helping the formation and maintenance of NDRs at TSS [3]. The DNA sequence surrounding SHL2, including the presence of poly-AT tracts, can also influence Chd1 activity, with the potential to establish a rate-limiting step by slowing down sliding [125]. Finally, the H2A.Z exchange rate by SWR1-C can be modulated by the nucleosomal DNA sequence [46].

Notably, the lobes of Chd1 and ISWI establish contacts with the second DNA gyre, transiently guiding DNA translocation during the conformational cycle [126,127]. Consistently, the lobe 1 of Sth1, the ATPase of the yeast SWI/SNF-subfamily remodeler RSC, also binds the second gyre, around SHL-6 [75], and the ATP-bound closed conformation of SWR1 displays increased contacts of lobe 1 with the second DNA gyre [66] (Figure 4).

#### 4.5.6. Impact of Histone Post-Translational Modifications and Variants

All remodelers contain subunits and/or domains that act as ‘readers’ of histone PTMs with specific affinities, potentially affecting both their targeting and regulation (Figure 4). Examples of readers include bromodomains (BRD), bromo-adjacent homology (BAH) domains, chromodomains (CHDs), plant homeodomains (PHD), Pro–Trp–Trp–Pro (PWWP) domains, and tryptophan-aspartic acid (WD40) domains. Conceptually, the presence of modifications (or a histone variant) can positively or negatively affect the targeting or regulation of a remodeler. However, due to their weak affinities, histone modifications are predicted to work in combination with each other or collaborate with additional factors in order to affect targeting. In SWI/SNF-subfamily remodelers, the histone-binding lobe of the substrate recruitment module clusters many BRDs (from Rsc1 or Rsc2 and Rsc4 in RSC; from BAF180 in PBAF) and appears to be poised to bind histone PTMs [73]. Notably, the histone-binding lobe does not exist in BAF, and might be specific to RSC/PBAF remodelers.

Examples of PTMs targeting the remodelers abound for all subfamilies of remodelers. In the ISWI subfamily, a PHD domain in the NURF remodeler and a PWWP domain in the ISW1b remodeler recognize H3K4me3 and H3K36me3, respectively [128,129,130]. In the CHD subfamily, CHDs in human CHD1 and CHD4 bind methylated histones [131,132]. Moreover, H3K9 acetylation and methylation both can independently enhance the association of the nucleosome with the PHD fingers of CHD3, the motor subunit of the NuRD remodeler [133]. In the SWI/SNF subfamily, the BRD in the Snf2 ATPase promotes its targeting to nucleosomes acetylated on histone H3 [134]. Similarly, the tandem BRDs located in Rsc4 display specificity for H3K14ac, enhancing binding of the RSC remodeler to nucleosomes [135,136] and the RSC remodeling activities are regulated in part by histone PTMs [100]. Remarkably and counter-intuitively, by enhancing the interaction between RSC and the nucleosome, acetylation of the +1 nucleosome restrains RSC remodeling activity to one H2A/H2B dimer eviction [137,138,139]. This explains why, while bound by a SWI/SNF-subfamily remodeler, the +1 nucleosome persists during the process of transcription initiation. Interestingly, the tandem PHD finger of human DPF3b, a component of the BAF remodeler, specifically recognizes H3K14ac and is deterred by H3K4me [140]. Moreover, the tandem PHD finger of human DPF2, an alternative paralog, selectively recognizes crotonylation at H3K14 [141], demonstrating extremely high stringency and specificity, and calling for further investigations. In the INO80 subfamily, the BRD-containing Bdf1 subunit promotes H2A.Z deposition by SWR1C on H4 or H2A acetylated nucleosomes [142,143].

Examples of PTMs regulating the remodelers abound as well. The stimulation of the ISWI ATPase and remodeling activities by the histone H4 tail basic patch is attenuated by H4K12ac or H4K16ac [89,100,102,144], while the latter modification does not affect spacing of nucleosome arrays [145]. Without altering affinity, site-specific H3 acetylation enhances remodeling activities of yeast RSC and SWI/SNF [100,146].

Some PTMs affect remodeling activities due to their location near the DNA entry site of the nucleosome, susceptible to significantly alter DNA affinity and breathing. Indeed, H3K56ac enhances nucleosome mobilization by SWI/SNF and RSC [147], and H3K64ac increases nucleosome sliding by Chd1 but not by RSC [148]. Notably, a remodeler can harbor a paralog that specializes it for remodeling partially unwrapped nucleosomes, as is the case with Rsc1-containing RSC [149]. Moreover, H3K56ac enhances the editing activity of INO80C, and remarkably alters the discrimination between H2A- and H2A.Z- containing nucleosomes by SWR1C [150], a result supported by the Arp5 ‘sensor toe’ binding to H3K56 [80].

Finally, PTMs can alter remodeling by weakening or interfering with the remodeler-nucleosome interaction. For example, ubiquitination at H2BK123 in yeast (K120 in human) sterically hinders the binding of the C-terminal tail/finger helix of Sfh1/Snf5/INI1-BAF47-SMARCB1 to the nucleosome acidic patch, likely reducing or abolishing nucleosome ejection by SWI/SNF-subfamily remodelers [73,75].

Histone variants can also impact remodeling. For example, H2A.Z-containing nucleosomes stimulate the ATPase and remodeling activities of ISWI, but not those of the SWI/SNF-subfamily remodelers, except in the context of nucleosome ejection in arrays [151,152]. Here, future work could explore how remodeling activities are influenced by combinations of specific PTMs and histone variants as observed in nucleosomes in vivo.

### 4.6. Regulation of Remodelers by Actin-Related Proteins Modules

Modules containing actin and/or actin-related proteins (ARPs) are heterodimers present in all large remodelers of SWI/SNF and INO80 subfamilies. They are also conserved from yeast to human and bind directly to the ATPase subunit. Overall, these modules play major regulatory roles in the alternative remodeling outcomes by regulating DNA translocation and/or nucleosomal DNA interaction.

One ARP module, present in the SWI/SNF-subfamily and INO80-subfamily remodelers, binds to the conserved HSA domain located upstream of the structural hub and ATPase domain. In yeast RSC and SWI/SNF remodelers, it associates Arp7 and Arp9 (along with Rtt102) and is implicated in the critical regulation of coupling, a measurement of DNA translocation efficiency, which drives the capability to achieve nucleosome ejection instead of sliding [34,35]. Moreover, the function of this HSA-bound ARP module is directly related to the regulatory role of the conserved structural hub, a physical multi-component bridge between the lobes, formed by the association of domains flanking and separating the lobes in the SWI/SNF-subfamily remodelers [14]. Here, biochemical and structural observations strongly support that the efficiency (the optimization of the conversion of the ATP binding and hydrolysis into DNA translocation), the synchronization of the lobes, and their grip on DNA are influenced by the position of the regulatory hub—which is guided itself by the position of the ARP module, and governed by the nucleosomal information collected by all the lobes of the substrate recognition module. Equivalent ARP modules combine ACTL6A and ACTB (along with BCL7A) in human BAF/PBAF/GBAF remodelers or incorporate Arp4 and Actin (along with Arp8) in yeast SWR1 and INO80 remodelers (Figure 4). In INO80C, in the absence of a nucleosome, this ARP module resides under the RuvBL1/2 ring [153]. In the current structures of the INO80-subfamily remodelers that also contain a nucleosome, the location of the ARP module is unfortunately not resolved, but is known via biochemical studies to increase the affinity of the remodeler for nucleosomes [84] and can act as an extranucleosomal DNA sensor [111,154,155]. Here, the role of the HSA-bound ARP module in sensing extranucleosomal DNA in the INO80-subfamily remodelers deserves further functional investigation, and in particular, the possibility that this module informs and regulates the lobes through a process similar to the one uncovered within the SWI/SNF-subfamily remodelers.

A second ARP module, present only in the INO80-subfamily remodelers, binds to the large insertion domain located between the two lobes of the ATPase domain. It combines Arp6 with Swc6 in the yeast SWR1 remodeler, and Arp5 with Ies6 in the yeast INO80 remodeler. In SWR1, this module functionally collaborates with Swc2, and the loss of Swc2 or Arp6 impairs histone exchange [41,47]. This module also tethers the remodeler to the histones and extends along the extranucleosomal DNA, leading to a substantial DNA unwrapping at the entry site [66]. In INO80, this module functionally collaborates with Ies2 to increase the affinity towards nucleosomes and also the ATPase activity while promoting coupling [84,85,86,87,156]. Structurally, this ARP module interacts with the proximal acidic patch, providing a histone anchor for the INO80-subfamily remodelers (Figure 4).

### 4.7. Regulation of Remodelers by Internal PTMs and Dimerization

Remodelers themselves can be modified by PTMs, adding another layer of opportunity for regulation. For example, internal phosphorylation or PARylation (i.e., the addition of polymers of ADP-ribose) reduces remodeling activities of human BAF and *Drosophila* ISWI, respectively [157,158,159]. Conversely, the CSB remodeler requires ubiquitination of its C-terminal region for most of its functions [160]. Interestingly, acetylation marks can play a switching role as they can be deposited on either the remodeler or its substrate. Indeed, the BRD of the Snf2 ATPase can interact with an acetylated residue internal to the ATPase, providing an alternative to binding acetylated H3 [161]. A very similar mechanism occurs for the BRD of Rsc4, which can interact with acetylated H3K14 or an internally acetylated residue [136]. Regulation by internal PTMs in remodelers are likely to be currently underappreciated and need further investigation, offering avenues for future studies.

Finally, functional dimerization might contribute to allosteric regulation of the remodelers, as characterized for SNF2H [71,162,163], but is not mechanistically required, per se, by the remodeler to achieve nucleosome sliding. Of note, the INO80 remodeler can also perform nucleosome sliding and may function as a dimer [51].

### 4.8. Regulation of Remodelers by Other Chromatin Components

Remodeling activities can also be regulated by their interaction with chromatin factors, possibly in cooperation with PTMs. For example, the recruitment of the SWI/SNF remodeler to gene promoters can be assisted by the activity of SAGA, a chromatin-modifying complex. Importantly, beyond targeting, transcription factors can regulate remodelers and influence outcomes. For example, the interaction between yeast SWI/SNF and a DNA-bound activator enhances nucleosome eviction [164,165,166]. Here, the modular composition of remodelers with cell-type-specific components might promote specific interactions with cell-type-specific activators and repressors, allowing distinct and highly tailored regulation.

Moreover, cellular differentiation often involves lineage-specific transcription factors that recruit specific remodelers, sometimes sequentially. For example, during muscle differentiation, the Transcription Factor (TF) MyoD first interacts with CHD2 to guide the deposition of H3.3 histone variant, and then with BAF to open the chromatin for transcription initiation [30,167]. In contrast, BCL6, the master regulator of B cell differentiation, recruits the NuRD remodeler and its repressive activity to prevent terminal differentiation into plasma cells [168].

Finally, the binding of multiple transcriptional activators to nucleosomal DNA at cis-regulatory elements (CREs) is inherently cooperative and can be critical to reaching a threshold response and a pattern in gene expression [169]. Interestingly, recent data from an in vivo quantitative detection of multiple transcription factors (TFs) at CREs measured genome-wide at the single-molecule level suggest that increased TF co-occupancy and cooperativity are required, but might not be sufficient, to open chromatin at sites of competition with nucleosomes, and that remodelers might be involved in TF co-occupancy [170]. The interplay between remodelers and TFs is of the utmost interest, and investigating the functions of accessory subunits of large remodelers in this context would be greatly beneficial.

## 5. Remodelers and Nucleosome Positioning at Enhancers and Promoters in Transcription Regulation

While nucleosome presence, composition, and positioning—and, thus, chromatin remodelers—play important roles in DNA replication, DNA repair, and DNA recombination, I will only focus on the contributions of remodelers to transcription regulation in this review. Globally, nucleosome positioning and chromatin organization result from the coordinated synergistic and antagonistic contributions of several remodelers in conjunction with many other chromatin factors. From their specialization, while ISWI and CHD subfamilies of remodelers are mainly involved in establishing and maintaining properly spaced nucleosomes, the SWI/SNF and INO80 subfamilies of remodelers are heavily involved in the formation and preservation of breaches into regularly spaced nucleosomes, named nucleosome-depleted regions (NDRs), particularly located at critical DNA regulatory elements, such as promoters and enhancers. Consistent with their functional specialization and contribution to specific chromatin landscapes, mammalian ISWI and SWI/SNF remodelers selectively mediate the binding of distinct transcription factors [10,171].

### 5.1. Roles of Remodelers at Enhancers

While working closely with promoters in mammalian cells, enhancers are also subject to the control of their accessibility by remodelers, exemplifying promoter-independent transcriptional regulatory roles for remodelers. For example, genome-wide, BAF remodelers maintain lineage-specific enhancers, regulate accessibility to distant enhancer sites, and actively contribute to tissue-specific gene activation [172,173,174]. BAF remodelers and their key components are also critical to proper stem cell self-renewal and pluripotency, as well as proper cell differentiation [175,176,177].

Thus, evidently, alterations in BAF remodelers perturb adequate accessibility of the transcription factors to the enhancers, disturbing cellular identity and differentiation programs [173,174,178,179,180,181,182]. Similarly, loss of CHD1 impacts the binding of androgen receptors (AR) at lineage-specific enhancers, driving prostate tumorigenesis and resistance to AR-targeted therapy [183,184].

Overall, it will be interesting to characterize how the action of remodelers at enhancers might synchronize with their action at promoters, and also how remodelers help in the commissioning of new enhancers, and the decommissioning of current enhancers, during differentiation.

### 5.2. Roles of Remodelers in the Regulation of Transcription by RNAPII

Globally, RNAPII promoters contain a combination of features mixed from two contrasting types of promoters based on their initial status: open or closed promoters [185,186] (Figure 5).

Open promoters are highly structured and accompany the most responsive genes. They are repressed but poised for transcription; in metazoans, they often contain pre-loaded paused RNAPII [187]. Importantly, they are characterized by clear NDRs, presenting accessible binding sites for transcription factors, resulting from DNA sequences (as AT-rich tracts) disfavoring nucleosomes [188] and active nucleosome ejection by SWI/SNF remodelers. They also present H2A.Z-containing +1 nucleosomes installed by SWR1 remodelers at uniform positions relative to the TSS, along with phased and regularly spaced nucleosomes [189] (Figure 5A, left). When the NDRs are wide enough (>150 bp), a common occurrence at open promoters, they might not be truly nucleosome-free, and instead, may harbor highly dynamic nucleosomes, termed ‘fragile’ nucleosomes [190,191,192,193] (Figure 5A, right). Fragile nucleosomes appear to be conserved as they have been identified in promoters of highly expressed genes in many organisms, for example, C. *elegans* [194], *Drosophila* embryos [195], and mouse Embryonic Stem Cells (ESCs) [196], and their presence and stability relate to the rate of transcription [197]. Importantly, in yeast, ‘fragile’ nucleosomes appear to be partially unwrapped nucleosomes resulting from the action of the SWI/SNF-subfamily remodeler RSC and subsequent binding of ‘general regulatory factors’ (GRFs) [198]. Here, Rsc1-containing RSC specializes in remodeling partially unwrapped nucleosomes, a feature common to NDR fragile nucleosomes and to tDNAs (i.e., transfer RNA-encoding genes) often flanked by highly enriched AT-rich sequences [149]. The activation of these promoters also relies on corresponding enhancers being open, increased histone acetylation, and further loss or repositioning of the nucleosomes around the TSS and beyond in the coding regions.

Closed promoters are transcriptionally repressed by the presence of nucleosomes covering the TSS and most TF binding sites. They are characterized by an absence of constitutive NDRs, AT-rich tracts, and pre-loaded RNAPII and display a low level of H2A.Z-containing nucleosomes (Figure 5B). Moreover, closed promoters are more likely to present a TATA box covered by a nucleosome, at least partially. Their activation is thus significantly dependent on remodelers changing the chromatin landscape locally. Overall, the degree of activation of these stably architectured promoters relies on the regulated opening of their chromatin through several concomitant or sequential chromatin transitions and remodeling events involving a pioneer activator, chromatin modifying complexes, chromatin remodelers specialized in chromatin opening by nucleosome ejection, and additional transcription activators. Interestingly, pioneer activators can recognize their cognate sites wrapped within a nucleosome prior to any nucleosome alteration and even aid remodeler recruitment. Furthermore, the initiation of transcription requires the action of remodelers specialized in chromatin opening. These remodelers antagonize the repression by histones [199], facilitate TBP binding [200], interact with a wide variety of activators depending on the functional context (e.g., ySwi5 or yGcn4p for ySWI/SNF [201,202]; HSF1 or the glucocorticoid receptor (GR) for hSWI/SNF [203,204]; dGAGA, dHSF, the ecdysone receptor, dTrf2, and the dKen repressor for dNURF [205,206,207]; and YY1 for hINO80 remodeler [208,209]), and render key promoter elements accessible by clearing promoters from nucleosomes and strengthening the NDRs (as done by ySWI/SNF [210]; yChd1 [28]; INO80 [211].

More acutely, at closed promoters, various remodelers from different families can act antagonistically, leading to a dynamic variation between opening/disassembly and closure/assembly, ultimately resulting in gene activation or repression, respectively [212]. This balance is of the utmost importance for tuning gene expression by transiently creating and maintaining an NDR, as well as determining the positioning, composition, and integrity of the +1 nucleosome. For example, in yeast, RSC and ISW1a act antagonistically; this is particularly visible at closed promoters where the loss of RSC leads to a limited fill-in of nucleosomes in the NDR of many RNAPII genes with a shift of the +1 and subsequent nucleosomes toward the NDR partially resulting from the ISW1 remodeler activity [213].

In yeast, the presence of specific DNA motifs influences the formation and maintenance of the NDRs [3,214]. Strikingly, binding and activity of the RSC remodeler at the NDRs depend on a pair of specific DNA motifs: a poly(A) tract, and a CGCG-containing sequence preferentially bound by the Rsc3 subunit [213,215,216]. Consequently, RSC contributes to the maintenance of promoter accessibility by actively excluding nucleosomes from NDRs via two approaches: sliding the +1 nucleosomes toward the ORF and ejecting nucleosomes to form large NDRs [193,198,217,218]. In addition to remodelers, pioneer activators and the GRFs destabilize nucleosomes in the vicinity of their cognate sites [215,219], contributing to the generation and maintenance of NDRs at open promoters. Here the RSC remodeler and the GRFs act independently but coordinate to prevent the occlusion of the TATA box by sliding the +1 nucleosome [218]. Aside from RSC, ISW1a, ISW2 and INO80 remodelers are also involved in the +1 nucleosome positioning [213,220,221]. Based on a reconstituted system using purified components, RSC, along with GRFs, performs DNA sequence-driven directional NDR-widening, and the +1 nucleosome positioning can be set by the cooperative action of GRFs with ISW1a and ISW2, or INO80 alone [214]. Moreover, with the +1 nucleosome acting as a reference, ISW2 and INO80 cause the distribution of nucleosomes in the ORF while proper spacing is tuned by the action of the ISW1a remodeler along with Chd1 [214,222].

During transcription, the passage of RNAPII through nucleosomes requires the displacement of the H2A/H2B dimer, which protrudes from the surface of the nucleosomal disc [107,223,224]. The latest structures propose and strengthen an elegant model in which the Chd1 remodeler assists the progression of the RNAPII by translocating DNA towards the polymerase while exposing the proximal H2A/H2B dimer for removal by the FACT chaperone in a processive mechanism [107,108,225]. Of note, TSS-proximal nucleosomes also present a higher cyclizability on the promoter-proximal face than the distal face, maybe favoring RNAPII progression [11].

Aside from their critical roles in promoter regulation during transcription initiation and assistance to RNAPII progression during elongation, remodelers also maintain and/or restore chromatin integrity in the gene body [130] and prevent antisense transcription [220] and initiation from cryptic intragenic promoters [226].

In counter distinction, but following similar principles, the repressed state of chromatin is maintained by the constant work of chromatin modifiers and remodelers that interact with DNA-bound repressors [227], slide nucleosomes over key promoter elements [228], prevent cryptic antisense transcription from intergenic regions [220], prevent TBP binding [229,230], and limit the size of the NDRs [231]. Notably, some remodelers involved in repression, such as NuRD, have embedded in their composition a histone deacetylase enzyme (HDAC) and methyl-binding proteins (MBDs), which intrinsically coordinate activities to achieve gene repression specifically at DNA-methylated regions.

### 5.3. Roles of Remodelers in the Regulation of Transcription by RNAPI and RNAPIII

The transcription of rDNA genes by RNA Polymerases I (RNAPI) is tightly regulated through nucleosome positioning, which is dynamically established by several remodelers. Silencing of rDNA genes relies on the relocation of the promoter-bound nucleosome to a position unfavorable for transcription through the action of the NoRC remodeler recruited by the TTF-I TF [232,233,234]. In addition, the histone variant H2A.X deposited at rDNA promoters in ESCs contributes to the recruitment of NoRC, thereby repressing rDNA transcription and limiting proliferation [235]. The NuRD remodeler also helps poise rDNA genes for transcription activation by contributing to a specific chromatin landscape [236]. Remarkably, NuRD shifts the position of the promoter nucleosome to the transcriptional off position upon direct interaction of its ATPase CHD4 with the long noncoding RNA PAPAS, which is upregulated in a stress-dependent manner and forms a DNA–RNA triplex structure at rDNA enhancers [237,238]. The activation of the rDNA genes involves counteracting remodelers such as the B-WICH remodeler, which responds to glucose [239,240], and the CSB remodeler, which resets the promoter-bound nucleosome position and enables transcription [241]. Of note, the α-thalassemia X-linked mental retardation (ATRX) remodeler localizes to rDNA during metaphase [242] and the deletion of CHD7 has been reported to induce aberrant rDNA silencing [243]. Finally, in yeast, the remodelers Isw2 and Ino80 actively contribute to the transcription of a fraction of the 35S ribosomal RNA genes and to the positioning of the nucleosomes flanking the ribosomal origin of replication [244]. Of note, like in the regulation of transcription by RNAPII, GRFs also contribute to the regulation of ribosome genes [245].

Finally, the transcription of small ncRNAs by RNA Polymerases III (RNAPIII) depends on the removal of nucleosomes by the SWI/SNF-family remodeler RSC in yeast [217], but more work is needed in humans to functionally characterize remodelers present at and assisting expression of ncRNAs.

## 6. Dissonances in Chromatin Remodeling in Cancer

Conceptually, in the process of oncogenesis, an initial epigenetic misregulation providing a cellular advantage, or enabling the tolerance to additional genetic mutations, can be selected and, through iterations, lead to a progressive acquisition of an oncogenic or metastatic state. Alternatively, epigenetic misregulation can lead to the maintenance of, or the transition to, a poorly differentiated transit-amplifying cellular state. Here, an initial, or subsequent, epigenetic misregulation can arise from dissonances in chromatin remodeling; and chromatin remodeler activities can be altered in various ways in cancer: redistribution or mistargeting; reduced or excessive subunit expression (potentially leading to rogue residual complexes); loss-of-function mutations in SWI/SNF-subfamily remodelers potentially leading to abrogation/decrease of remodeling and DNA accessibility at promoters and enhancers of tumor-suppressor or other genes (and the converse with loss-of-function mutations in assembly remodelers); or gain-of-function mutations in SWI/SNF-subfamily remodelers potentially leading to increases in nucleosome mobility and DNA accessibility at oncogenes or genome-wide. Notably, while mutations in remodelers from the SWI/SNF subfamily are present in >20% of human cancers (see Section 6.2), various remodelers from all subfamilies are increasingly being implicated in cancers.

### 6.1. Alterations of ISWI-, CHD-, and INO80-Subfamily Remodelers in Cancer

In the ISWI-subfamily remodelers, the ATPase of the human NURF remodeler SNF2L attenuates Wnt/β-catenin signaling, suppressing cell proliferation and migration. Remarkably, SNF2L is almost absent in melanoma cells while being robustly expressed in normal melanocytes [246]. Interestingly, in *Drosophila*, the dNURF complex regulates larval blood cell development, and a deficiency of dNURF leads to a neoplastic transformation of circulating hemocytes, resulting in blood cell overproliferation and melanotic tumors [206]. High levels of SNF2H are necessary for intensive cell proliferation and cell cycle progression of developing hematopoietic stem cells (HSCs) and for completion of erythropoiesis in mice [247]. Interestingly, in acute myeloid leukemia (AML) patients, CD34+ hematopoietic progenitors show SNF2H upregulation [248], which can be drug-inhibited to release terminal-differentiation while sparing normal hematopoiesis [249].

In the CHD-subfamily remodelers, loss of CHD1 impacts the binding of androgen receptors (AR) at lineage-specific enhancers, driving prostate tumorigenesis and resistance to AR-targeted therapy [183,184]. Deficiency in CHD2, which is essential for proper hematopoietic stem cell differentiation, leads to lymphoma [250]. Being the major gene-repressing remodeler, the CHD4-containing NuRD complex is the CHD-subfamily remodeler with the most connections to cancer. The MTA1-3 components of NuRD regulate invasive behavior in several cancers, with unique and often antagonistic activities. For example, tumor progression in many cancer types is observed when MTA1 expression is increased, while MTA3 limits breast tumor progression by repressing a master regulator [251,252,253]. Via its associated MBD proteins, such as MBD2, NuRD can also be mistargeted by aberrant DNA methylation commonly occurring in cancer cells, supporting tumorigenesis by silencing tumor suppressor genes (TSGs) [254,255]. Remarkably, CHD4 can even play an upstream role by initiating abnormal de novo DNA methylation, promoting the maintenance of TSGs silencing and thus colorectal cancer cell proliferation, invasion, and metastases [256]. In endometrial cancer, CHD4 depletion by specific hot-spot missense mutations promotes tumorigenesis by increasing cancer stem cell characters through the TGFβ signaling pathway [257]. An elegant in-depth mechanistic analysis of CHD4, using cancer-associated missense mutations transposed into the *Drosophila* homologue Mi-2, revealed heterogeneous defects (i.e., reduction in protein stability, disruption of DNA binding, and loss of ATPase activity or coupling), leading mainly to loss-of-function [258], nicely supported by structural work [72]. However, one CHD4 cancer-associated mutation located in the Brace-I helix (H1196Y) leads to a gain in remodeling efficiency [258], a result remarkably consistent with that of a similarly located mutation (K938A) introduced in BRG1/Sth1, a SWI/SNF-subfamily remodeler ATPase [14], strongly suggesting a conserved regulatory function for this region across remodelers from different subfamilies. Of note, the dedifferentiation of triple-negative breast cancer cells is driven in part by the activation of the NuRD remodeler via a MUC1-C (oncogenic mucin 1 C-terminal subunit)-MYC pathway [259]. CHD5, a paralog of CHD4 that is preferentially expressed in neural tissue and testis, and forms a NuRD-type remodeler [260], is a tumor suppressor involved in the regulation of genes related to neuroblastoma [261] and gliomas [262]; and the interaction of the PHD finger of CHD5 with an unmodified histone H3 tail is essential to restrain tumorous growth of neuroblastoma cells in vivo [263]. Moreover, in a variety of other malignancies, CHD5 emerged as a TSG with decreased expression, often resulting from one allele being deleted and the promoter of the other allele being silenced by hyper-methylation. Finally, CHD5 can be silenced in neuroblastomas by MYCN-driven miRNAs [264]. Remarkably, the CHD8 remodeler is linked to the transcriptional coactivator BRD4 via a short isoform of NSD3, likely to facilitate chromatin remodeling and transcription activation in AML cells [265].

In the INO80-subfamily remodelers, despite their extensive involvement in DNA repair and recombination, mutations appear to be uncommon in human cancers. However, INO80 plays an essential role in superenhancer-mediated oncogenic transcription and tumor growth, in both melanoma and non-small-cell lung cancer [266,267]. Interestingly, INO80 counteracts R-loops, promoting DNA replication in the presence of transcription, enabling proliferation in cancers [268].

### 6.2. Alterations of SWI/SNF-Subfamily Remodelers in Cancer

Mutations in subunits of SWI/SNF-subfamily remodelers are present in >20% of human cancers and are observed at high frequencies in malignant rhabdoid tumors (MRTs) (>95%), ovarian clear cell carcinoma (75%), clear cell renal carcinoma (57%), hepatocellular carcinoma (40%), gastric cancer (36%), melanoma (34%), and pancreatic cancer (26%) [269,270]. Importantly, in many cancers involving mutations in SWI/SNF components, few, if any, co-occurring genetic mutations are found, indicating that SWI/SNF mutations have the potential to be driver mutations by providing an advantage for tumor initiation or progression.

Mammalian cells contain three major SWI/SNF-subfamily remodelers: BAF (or cBAF, canonical BAF) [271], PBAF (polybromo-associated BAF) [272], and GBAF (or ncBAF, non-canonical BAF) [273,274]. All harbor an ATPase subunit, BRG1 or BRM (the latter not present in PBAF), along with core subunits BAF155, BAF53A/B, BAF60A/B/C, and actin. In addition to the core subunits, BAF uniquely contains ARID1A/B, BAF45B/C/D, DPF2, BCL7A/B/C, and BCL11A/B; PBAF uniquely contains ARID2, PBRM1, BAF45A, and BRD7; and GBAF uniquely contains GLTSCR1/GLTSCR1L paralogs and BRD9 [273,274]. Additionally, BAF170, BAF47, and BAF57 subunits are shared by BAF and PBAF, and the SS18/CREST subunit is shared by BAF and GBAF. Paralogous subunits can be expressed simultaneously, mixing unique and redundant functions in the same cell [275,276], but they can also be exchanged to accompany cellular differentiation and the establishment of a cell-type-specific transcriptional program [277,278]. Overall, beyond being located in functional hotspots affecting the catalytic activity and efficiency, many BAF cancer-associated mutations cluster at key structural interfaces between subunits or between the remodeler and the nucleosome, and attenuate remodeling activity [79], highlighting the relevance to further functionally characterize the composition of each remodeler and the conversations between remodelers and chromatin.

#### 6.2.1. Alterations of the ATPase Subunits

Alterations can affect the ATPase subunits, BRG1 and BRM, in the human BAF/PBAF/GBAF remodelers, with the BRG1 subunit containing the majority of cancer-associated missense mutations. Among them, the most common mutations affect residues located on the surface of the lobes and are involved in DNA binding or the formation of the ATP-binding pocket. They are functionally characterized as missense loss-of-function mutations as they lead to reduced or abrogated ATPase and remodeling activities, and decreased DNA accessibility [79,279,280,281]. In contrast, rare cancer mutations, located in a specific region of the conserved structural hub that bridges the two ATPase lobes, act as missense gain-of-function mutations as they increase DNA translocation efficiency, remodeling activities, and DNA accessibility [14].

Loss or reduced expression of BRG1 can drive tumorigenesis [279]. In contrast, overexpression of BRG1 has been observed in most human breast cancer tumors, and BRG1 knockdown has been shown to sensitize triple negative breast cancer cells to chemotherapy drugs [282]. The alternative ATPase subunit BRM leads to different transcription specificities, with a possible antagonistic function in differentiation [283,284]. Although Brm homozygous knockout mice present androgen-independent growth and cellular proliferation [285], there are fewer links of BRM, than BRG1, with cancer.

Importantly, synthetic lethality, the loss of cell fitness upon combining two genetic events, has been demonstrated, with the loss of BRG1 and the simultaneous inhibition of BRM [286,287,288]. Thus, there is an increased dependency upon a particular subunit paralog, which can be therapeutically targeted. However, 2% of the MRTs and specific tumors, such as small cell carcinoma of the ovary, hypercalcemic type (SCCOHT), and SMARCA4-deficient thoracic sarcomas (SMARCA4-DTS), present dual deficiencies of BRG1 and BRM via a genetic loss of BRG1 combined with the silencing of BRM [289,290]. Notably, although appealing, designing drugs to discriminate specifically between BRG1 and BRM is challenging, as targeting their bromodomains did not reveal a decisive vulnerability [291], and their ATPase domains are extremely similar.

#### 6.2.2. Alterations of Other Core or Accessory Subunits

Alterations can affect other core or accessory subunits of the BAF/PBAF/GBAF remodelers.

The vast majority of the MRTs (~98%) exhibits a biallelic loss of the BAF47 subunit (also named SMARCB1, INI1, and hSNF5) [292,293], and a loss of EZH2 subunit of the PRC2 complex prevents the formation of tumors in Snf5^−/−^ mice [294]. Moreover, specific EZH2 inhibition impacts BAF47-deficient tumors favorably [295,296]. While being dependent on BRG1, BAF47-mutated MRTs are sensitive to BRD9 chemical and biological depletion and inhibition [274,297,298,299]. As BRD9 is a GBAF-specific subunit, GBAF appears critical for the maintenance and proliferation of MRT cells, and its subunits can thus be synthetic lethal targets in MRTs. Similarly, AML can be dependent on SWI/SNF subunits, BRG1, BRD4, and BRD9, all components of GBAF, and exhibit sensitivity to BRD9 inhibition [300,301,302]. Remarkably, the C-terminal domain (CTD) of BAF47 interacts with the distal acidic patch of the nucleosome; and this interaction can be disrupted by mutations, rendering BAF unable to increase DNA accessibility [124]. This inability to increase DNA accessibility is consistent with that observed upon deletion of the yeast Sfh1 CTT domain (equivalent to the CTD of BAF47), which is required for nucleosome ejection but not ATPase activity or nucleosome sliding [73].

In parallel, synovial sarcoma is defined by the hallmark SS18–SSX fusion oncoprotein, which leads to a rare gain-of-function by mistargeting BAF and activates bivalent genes located at broad polycomb domains [303,304]. Remarkably, SSX in the SS18–SSX fusion protein binds to the acidic patch of H2AK119Ub-containing nucleosomes, recruiting mutated BAF to erroneous locations, and activating cancer-specific transcription programs [305]. Thus, alongside structural relevance of BAF47 discussed above, the lack of interaction (due to the lack of BAF47 in the context of MRTs) or the hijacking of the interaction with the distal nucleosomal acidic patch (due to the SS18–SSX fusion in the context of synovial sarcoma) can dramatically alter the regulation or the targeting of BAF, respectively.

Mutations in the BAF-specific ARID1A/B paralogs are frequent in melanoma, ovarian [306], gastric, and pancreatic cancers. Indeed, ARID1A is the most commonly mutated SWI/SNF subunit in cancer, altering transcriptional regulation that cannot be covered by the ARID1B paralog [178,179]. Nevertheless, due to functional redundancy at enhancers, ARID1B is remarkably essential in ARID1A-deleted cancers [178,307]. Evidently, these results demonstrate a synthetic lethality relationship between ARID1A and ARID1B, with a partial functional redundancy of paralogous subunits, signaling potential therapeutic targets. Of note, additional synthetic lethal targets have been identified outside BAF, within signaling pathways and other complexes, such as PI3K/AKT-pathway [308], EZH2 [309], PARP [310], and ATR [311], revealing vulnerabilities and therapeutic opportunities to explore. Additional synthetic lethalities between BAF subunits have been discovered in a systematic screen and can be explored for therapeutic approaches [312].

Notably, the functional specificity of the PBAF remodeler arises from its specific ARID2, PBRM1, and BRD7 subunits. PBRM1 is characterized by six bromodomains in tandem, two BAH domains, and an HMG domain [272,313], and mutations affecting this subunit have been identified in >40% of renal cell carcinoma [314]. Recently, synthetic lethality and apoptosis in PBRM1-deficient cancer cells have been induced by a specific EZH2 inhibitor [315]. Of note, BRD7, which also contains a bromodomain, is a tumor suppressor identified in a subset of breast cancers lacking p53 mutations [316,317]. Thus, the functional characterization of the multiple bromodomain containing PBAF-specific module is of great importance to aid the understanding of its alterations in the context of specific cancers.

Overall, the identification of potential vulnerabilities in cancer cells harboring mutations in remodeler genes has led to the intense ongoing exploration of a broad variety of therapeutic approaches targeting various remodeler subunits, chromatin regulatory complexes, and signaling pathways, all involved in the crafting of the chromatin landscape and the resulting cellular identity.

## 7. Final Remarks and Future Directions

Chromatin remodeling is the cornerstone of the dynamics occurring in the chromatin landscape during any DNA transaction. All remodelers harbor a conserved catalytic ATPase domain that converts ATP binding and hydrolysis into DNA translocation, the underlying mechanism for all nucleosome remodeling outcomes. Even the ways by which remodelers engage the nucleosome are highly conserved and coherent with their respective functions, displaying a necessary increase in complexity to judiciously overcome challenges specific to each remodeling process and outcome.

The recent bloom of multiple spectacular structures of remodelers bound to nucleosomes nicely substantiates the profusion of genetic, biochemical, and biophysical discoveries. Those structures also represent a leap forward with the detailed visualization of many subunits and the striking architecture of the large remodelers based on a backbone ATPase subunit spanning the entire remodeler from which critical modules arise to interact with specific nucleosomal features. Mechanistically, however, they only represent snapshots of a complex mechanism requiring significant conformational changes for sampling nucleosomes, validating the appropriate one as a substrate, engaging it, remodeling it adequately, and ultimately releasing it. It will be fascinating and highly informative to obtain structures from the different successive steps of the remodeling cycle. Additionally, for large remodelers from the INO80and SWI/SNF subfamilies, entire or substantial parts of subunits known to play critical roles in regulation and substrate recognition are currently missing. Among them, the PBAF-specific module containing multiple BRDs is structurally and functionally of great interest. Finally, it is appealing to structurally visualize and understand dysfunctional remodelers harboring detrimental mutations, for example, cancer-associated mutations, to potentially guide the development of highly specific therapeutics.

Moreover, nucleosomes are not isolated entities in vivo; thus, assessing further chromatin remodeling and the interactions between the remodeler and the nucleosome in its chromatin ecosystem with neighboring nucleosomes will be enlightening. Furthermore, remodelers work cooperatively with other key chromatin components, for example, histone chaperones, in all chromatin processes, and additional functional and structural work will help shed more light on these interplays.

Finally, the structural and functional collaborations between remodelers and their targeting and regulatory factors, such as pioneer transcription factors, activators, and repressors, are worth further investigation, particularly in the perspective of transcription regulation. Since all remodelers establish contacts with extranucleosomal DNA, it is valuable to investigate how specific DNA sequences, in the presence or absence of DNA-binding factor(s), inform and regulate the remodeler. Similarly, remodelers contain several PTM reader domains, and a closer investigation of their targeting and regulation resulting from collaborative bindings would be highly relevant. Thus, understanding the flow and integration of complementary information received by distinct functional modules of the remodelers from various components of the chromatin landscape, and the resulting adjustments in DNA translocation to achieve different outcomes, remains wide open for further investigations.

Overall, a deeper mechanistic, regulatory, and functional characterization of the conversations between specific chromatin landscapes and specialized remodelers will also contribute to the understanding of the dissonances occurring in cancer and the uncovering of innovative therapeutic strategies to explore.

## Figures and Tables

**Figure 1 ijms-22-05578-f001:**
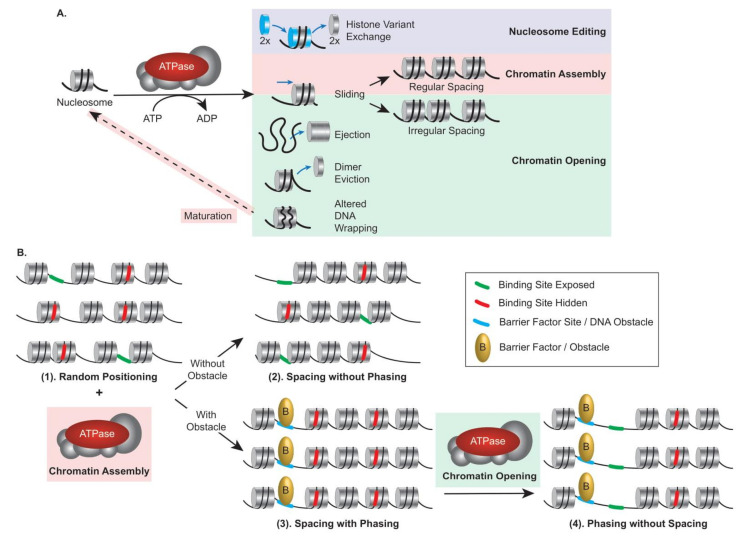
Chromatin processes, nucleosome spacing and phasing, and remodelers’ contribution. (**A**) Contribution of remodelers to various chromatin processes. Using their ATPase subunit, remodelers can contribute to (i) chromatin opening (light green background) by ejecting nucleosomes, irregularly spacing nucleosomes through sliding, evicting dimers, or altering DNA wrapping; (ii) chromatin assembly (light pink background) by maturating deposition or regularly spacing nucleosome; and (iii) nucleosome editing (light purple background) by modifying octamer composition through installation or removal of histone variants (blue partial cylinder). (**B**) Contribution of remodelers to nucleosome spacing and phasing. (1) Within a genome population, the deposition of histone octamers results in random nucleosome positioning with a mix of blockage (red) and exposure (green) of specific cognate sites for DNA-binding proteins. (2) Upon remodeling by an assembly remodeler, without an obstacle, nucleosomes are regularly distributed along the DNA, resulting in regular spacing, but without phasing, as access to particular cognate sites remains heterogeneous across the population. (3) Alternatively, upon remodeling by an assembly remodeler, in the presence of an obstacle, e.g., a DNA sequence or a bound barrier factor, nucleosomes are regularly distributed along the DNA, resulting in regular spacing and phasing with homogeneous access to particular cognate sites across the population. (4) From a population of regularly spaced and phased nucleosome arrays, upon remodeling by an opening remodeler, nucleosomes can remain phased but lack regular spacing, leading to a homogeneity in the exposure of binding sites across the genome population.

**Figure 2 ijms-22-05578-f002:**
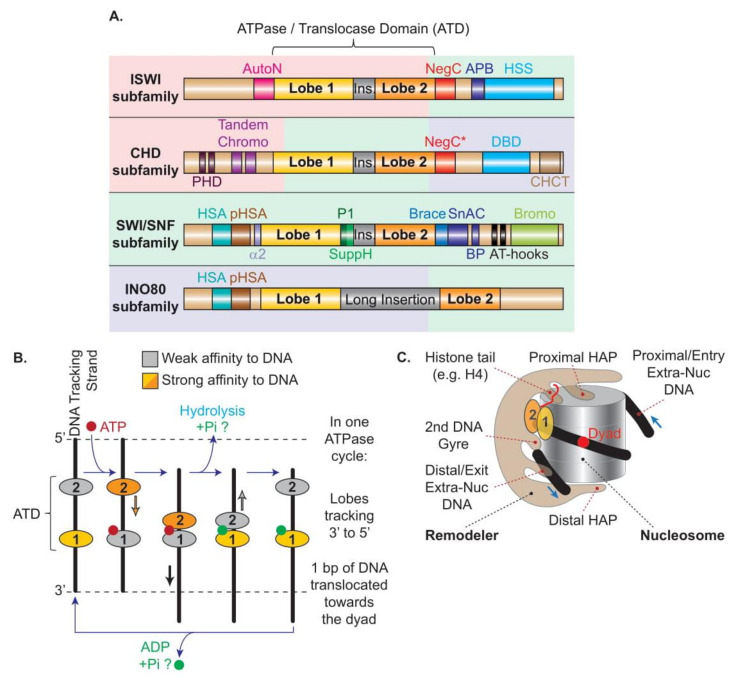
ATPase-based classification of remodelers, model for DNA translocation, and mechanistic and regulatory contacts between a remodeler and a nucleosome. (**A**) ATPase-based classification of remodelers. All remodelers contain a SWI2/SNF2-related ATPase subunit characterized by an ATPase/translocase domain (ATD), which is split into two RecA-like lobes, lobe 1 (yellow) and lobe 2 (orange), and sufficient to perform DNA translocation. Remodelers can be separated into four subfamilies based on the conserved domains flanking the ATPase domain and the length of the insertions between the lobes (gray). The INO80 subfamily is the only subfamily bearing a long insertion. Types and position of domains further define each subfamily. In SWI/SNF, an HSA helix, a post-HSA, an α2 helix, a SuppH-containing Protrusion 1 helix (P1), and a Brace that together form a structural hub bridging the lobes; a SnAC domain along with a basic-patch (BP); AT-hooks; and a bromodomain. In ISWI, an AutoN region, NegC region, APB domain, and HSS (HAND–SANT–SLIDE) module. In CHD, possible PHD fingers, tandem chromodomains, a region similar to NegC, and a DBD (DNA-binding domain). In INO80, an HSA helix and a post-HSA. Possible remodeling outcomes from each subfamily are color-coded in the background of each ATPase as in Figure 1A. (**B**) Model for DNA translocation. The lobes of the ATPase domain (depicted as in (A)) undergo an ATP binding- and hydrolysis-dependent conformational cycle that correlates with alternating high affinity for DNA, driving DNA translocation. Lobes are colored (yellow for lobe 1; orange for lobe 2) when they have a high affinity for DNA and depicted in gray when they have a low affinity for DNA. Only the tracking strand of DNA, along which the lobes move in a 3′–5′ direction, is represented. The movements of the lobes are visualized by colored arrows, and the DNA translocation is depicted with back arrows. The precise step in which the inorganic phosphate (Pi) is released is unknown. Model inspired from Reference [13] and modified to incorporate results and observations from References [14,15], leading to an updated model in which lobe 1 is the stationary lobe. (**C**) Mechanistic and regulatory contacts between remodelers and nucleosomes. Beyond the lobes (here depicted at SHL2 from the dyad), depending on the subfamily, multiple mechanistic and regulatory contacts can be established between a remodeler (brown shape) and a nucleosome (cylinder with wrapped DNA in black; dyad axis depicted). The ATPase itself or additional subunits interact with (1) the second DNA gyre; (2) the extranucleosomal DNA on the entry or exit side; (3) the proximal and/or distal histone acidic patches (HAP); and (4) the histone tails, for example, H4 tail depicted here (red). Direction of potential translocation is indicated by blue arrows.

**Figure 3 ijms-22-05578-f003:**
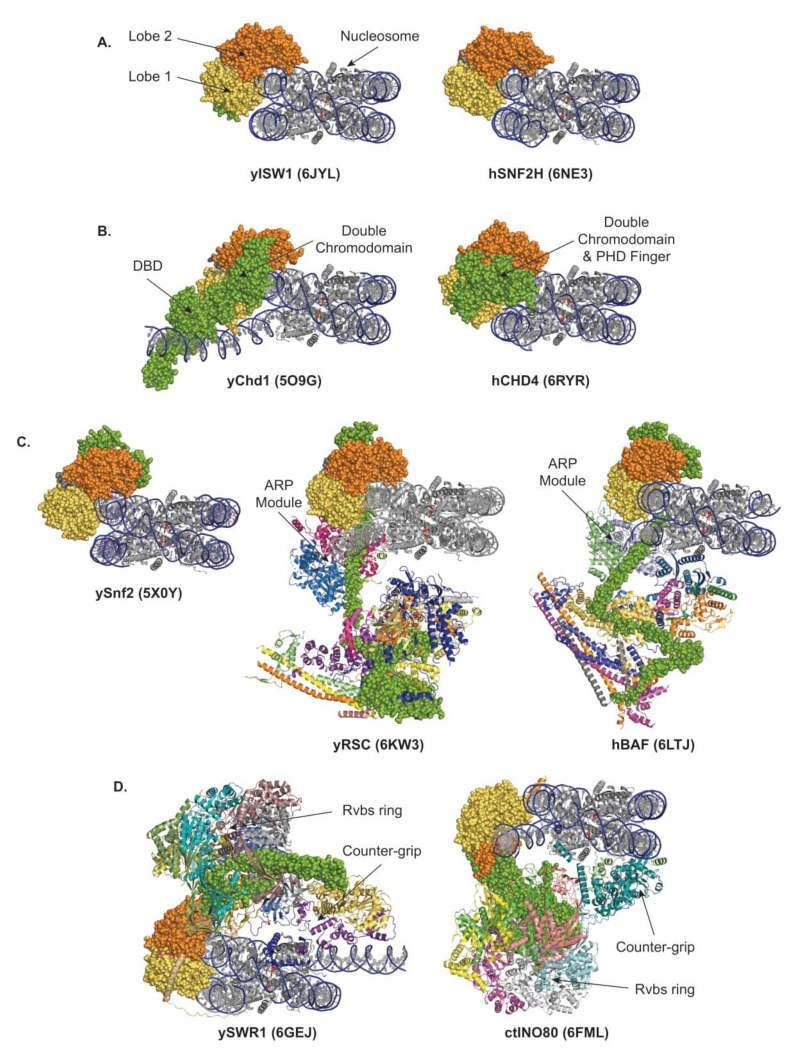
Cryo-EM structures of nucleosome-bound remodelers. Remodelers from the (**A**) ISWI subfamily are yeast Isw1 (6JYL), and human SNF2H (6NE3); (**B**) CHD subfamily are yeast Chd1 (5O9G) and human CHD4 (6RYR); (**C**) SWI/SNF subfamily are yeast Snf2 (5X0Y), yeast RSC (6KW3), and human BAF (6LTJ); and (**D**) INO80 subfamily are yeast Swr1 (6GEJ) and *C. thermophilum* Ino80 (6FML) bound to a nucleosome are depicted in the same orientation resulting from a structural alignment of their respective histone octamers before being separated into panels. Corresponding features are depicted with the same color codes. The ATPase subunits are depicted in spheres, with lobe 1 in yellow, lobe 2 in orange (as in Figure 2), and the rest of the subunit in green. The other subunits, if any, are depicted as colored cartoons. The dyad base-pair of each nucleosome is depicted in red. A few remarkable features have been named and indicated by arrows. All the remodelers from the various subfamilies engage the nucleosome in a similar or an inverted manner.

**Figure 4 ijms-22-05578-f004:**
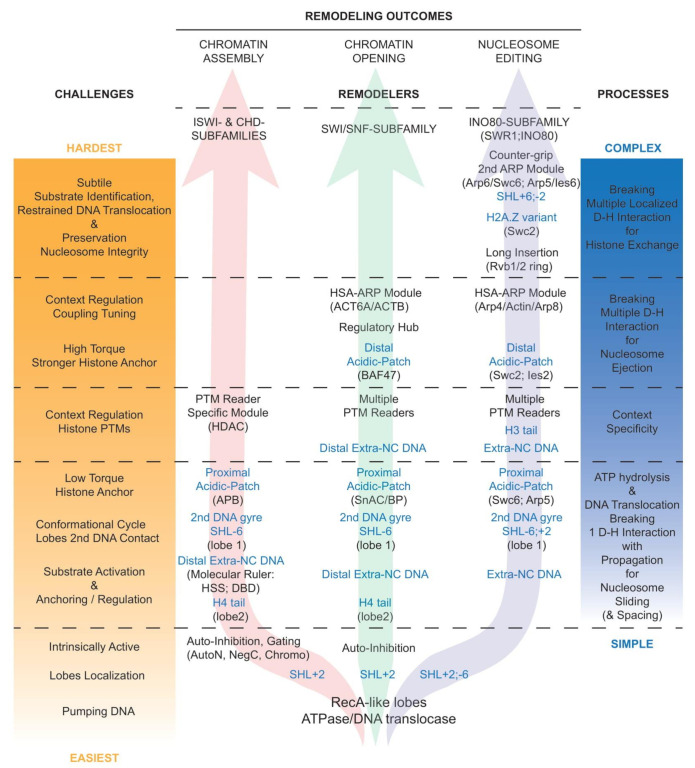
Hand-in-hand progressive increase in complexity of the regulatory conversations between remodelers and nucleosomes with the increase in challenges of remodeling processes. On the way (arrows) to successfully achieve their respective remodeling outcomes (top), remodelers from each subfamily need to overcome challenges (left orange gradient) associated with the processes they carry out (right blue gradient) by using shared and specific subunits and domains (black text along the arrows) corresponding to distinct nucleosomal features (blue text along the arrows). There is a positive correlation: the more complex the process is, the more challenges need to be overcome, and the higher is the need for complex regulation and multiple contacts with the nucleosome, and thus the more structurally and compositionally complex the remodelers are. D–H interaction stands for DNA–histone interaction.

**Figure 5 ijms-22-05578-f005:**
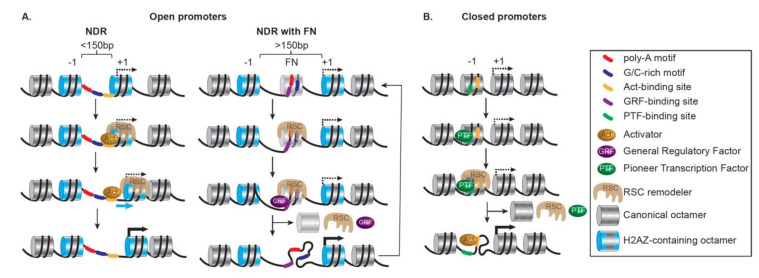
Promoter architectures and nucleosome occupancy impact how remodelers are recruited. For adequate regulation, most genes in yeast blend features from two types of promoter architectures. (**A**) Open promoters, in their repressed state, present an NDR without (left) or with (right) a fragile nucleosome (FN) adjacent to the TSS (black arrow), containing poly-A and G/C-rich motifs, along with activator/GRF-binding sites, and flanked by well-positioned H2A.Z-containing nucleosomes, leading to regular spacing and phasing in a genome population. Open promoters are common at constitutively active genes, and their activation necessitates minor (if any) remodeling (+1 nucleosome sliding or FN ejection) through direct remodeler recruitment by an activator or a GRF. (**B**) Closed promoters, in their repressed state, contain a continuum of canonical nucleosomes covering their TSS, leading to a genome population lacking phasing. Closed promoters are common at highly regulated genes, and their activation necessitates the binding of a pioneer transcription factor (PTF) in the nucleosomal context, followed by the recruitment of a remodeler, which opens the promoter by ejecting nucleosomes, rendering the activator-binding site accessible.

## Data Availability

Not applicable.

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
