# Peer review of "Sophisticated Conversations between Chromatin and Chromatin Remodelers, and Dissonances in Cancer"

_ijms, 2021, doi:10.3390/ijms22115578_

Round 1
Reviewer 1 Report
In the review article entitled “Sophisticated Conversations Between Chromatin and Chromatin Remodelers, and Dissonances in Cancer” by Cedric Clapier a current review on Chromatin remodelers is presented. Overall the review article is well written and structured and covers recent findings as well as to some extend historic perspective in the field of nucleosome structure and positioning by chromatin remodelers with an emphasis on transcription and cancer associated with defects in chromatin remodelers. This review article is of substantial comprehensiveness and provides detailed knowledge in many aspects of chromatin remodelling, such as the positioning of nucleosomes, structural aspects of chromatin remodelers and enzymatic steps required for the three major functions of chromatin remodelers. At some points the author could improve the article especially for readers not so familiar with the topic (see below).
Minor points:
Line 60: “… DNA methylation also alters DNA flexibility…” how is DNA flexibility affected?
Line 65 and many other occurrences in the manuscript: the author points at some other chapters in the manuscript by using “see below”. It would be easier for the reader if the author would provide the chapter that he is refereeing to. E.g. “see chapter 2.2.1”
Line 116: “e,g,” should read “e.g.”
Line 122 and ff.: I am missing a general description of the chromatin remodeler complexes as multi-factorial complexes. Especially with readers new to the filed in mind, I would suggest to add a description in section 2 highlighting the modular structure of chromatin remodelers and their basic principle in addition to the detailed description provided in section 2.2
Figure 3: I would suggest to structure the four classes of remodelers by adding either lines between or boxes around each class. It should also be added to the figure caption that the structures are based on cryo EM studies.
Line 466 and ff.: the references to figure 4 “(Figure 4)” are rather unspecific. I think it is not clear to what the author is referring. Is it possible the figure numbering is erroneous?
Line 496 and multiple other locations: PHD domain is spelled differently at different locations: e.g . “Plant HomeoDomain” vs “plant-homeodomains”. This should be unified.
Line 689: “PARylation” should be explained briefly.
Line 756: “alterations in BAF remodelers…”, what does alterations mean?
Line 788 “tDNA” should be explained
Figure 5: The contrast in some parts of the figure is low. E.g. RSC and Act is hardly readable.
Line 920 and 946: the Greek letter beta is not reproduced correctly.
Line 1141 and 1597: The first reference has “[see comments]”. It is not clear to what this refers.
Reviewer 2 Report
In the review “Sophisticated Conversations between Chromatin and Chromatin-Remodelers, and Dissonances in Cancer,” Clapier provides a thorough and detailed discussion of the current understanding of chromatin remodelers organizing the eukaryotic genome as well as in gene expression, and how anomalies to these processes arise specifically in cancer. Clapier introduces how the structural features of nucleosomes contribute to stabilizing DNA and how remodelers play an active role in editing nucleosomes and further compacting or opening chromatin to influence its higher-order structure and function. Clapier covers the main classifications of remodelers in terms of their sequence features, their specific interactions with nucleosomal and extranucleosomal DNA, and how their function is regulated. A specific example of how remodelers contribute to genome function is illustrated for the case of transcription, where remodelers act on enhancer elements and also on promoters to influence polymerase activity. Finally, the review concludes with how epigenetic misregulation and/or mutations to remodelers can contribute to their aberrant activity, thereby underlying cancer.
Overall, this is a highly well-written piece that will serve as a great resource to the field, and I only have a few minor points that would need to be addressed for publication.
Comments:
- The discussion in paragraph spanning lines 54-64 would benefit with inclusion of Ref. 11 (Basu Nature 2021) to emphasize how sequence-specific features of DNA influence its mechanical properties of
- This review would be improved by more explicitly defining or introducing the concept of “phasing” in terms of nucleosomal positioning, particularly in section 1.2
- The evolutionary conservation of specific internucleosomal spacing, biased towards 10n+5 bp versus 10 n, should be included in the context of how spacing is regulated (section 1.2). See Lohr and Van Holde, PNAS 1979.
- In terms of organization of the review, it may be helpful for the new reader if the review first introduce section 3.1, which outlines general features of remodelers, before section 2.2, which lists in great detail the specific features of several subfamilies of remodelers.
- Figure 4 is a difficult figure to interpret at first glance and is not intuitive with how or if the “challenges” are connected to the “processes” and how the sequence features relate to either the left/right hand column. For example, is low torque/histone anchor regulated by all the features or only proximal acid-path? What constitutes if a process is complex/simple or hard/easy – is there a quantitative basis for this metric or is everything qualitatively/arbitrarily ranked? Why is there no description of processes corresponding to “intrinsically active, lobes localization, pumping DNA” challenges? Overall, it is very unclear how the terms in each column relate to terms in the corresponding row, especially what is implied with the dashed lines. My advice would be to simplify the figure and include only one column corresponding to either challenges/processes and include the middle section. However, make it clear how the specific feature/modification of the current middle section is related to the challenge/process. Do not include the dashed lines, but use solid lines if there is a clear divide between the challenge/process and the associated feature/modification.
Minor comments:
- Several typographical errors:
- Line 31: “147 bp of DNA” [add a space between the number and unit]
- Line 36: “~1 pN” [add a space between the number and unit]
- Line 54: “depends on the underlying the sequence of DNA”
- Line 814: “Recruited.For”
- Sentence in Lines 82-84 is a bit of a run-on and not clear. Perhaps split it into sentences to help with clarity.
- Unfamiliar spiral symbol appears several times: Lines 150, 920, and 946
- The sentences in Lines 471-477 are lengthy with several inter-dependent clauses that are ambiguous as written. These should be reworded to be more direct/clear and not as circuitous.
- Section title 4.5.6 is on a separate page from the text (Line 585).
- Section 5.2 has some ambiguous wording. Line 767 “RNAPII promoters contain a combination of features mixed from two contrasting types of promoters depending on their initial status: open or closed promoters.” It’s unclear what there is referring to. The sentence makes it seem that a single RNAPII promoter is hybrid of open and closed promoters – or is the sentence implying that RNAPII promoters may be either open or closed? The wording seems contradictory in lines 770-1, especially when compared to the wording in lines 792-3: “Open promoters are highly structured and accompany the most responsive genes: they are thus repressed but poised for transcription” versus “Closed promotes are transcriptionally repressed by the presence of nucleosomes…”. Why would the most responsive genes “thus” have repressed promoters?
